



# Homogenization of the historical series from the Coimbra Magnetic Observatory, Portugal

Anna L. Morozova[1,2], Paulo Ribeiro[2], and M. Alexandra Pais[1,2]

[1] University of Coimbra, CITEUC, Department of Physics, Coimbra, Portugal
[2] University of Coimbra, CITEUC, OGAUC, Coimbra, Portugal

*Correspondence to*: A.L. Morozova (annamorozovauc@gmail.com)

**Abstract.** The Coimbra Magnetic Observatory (COI), Portugal, established in 1866, provides almost continuous records of the geomagnetic field elements for more than 150 years. However, during its long lifetime inevitable changes of the instruments, measurement procedures and even re-location of the Observatory took place. In our previous work (Morozova

et al., 2014) we performed homogenization – elimination of the artificial changes – of the measured declination series (D) for the period from 1866 to 2006. In this paper we continue work applying homogenization procedures to the measured series of the absolute monthly values of the horizontal (H, 1866-2006) vertical (Z, 1951-2006) and inclination components (I, 1866-1941). After homogenization of all measured series for the 1866-2006 time interval we performed the homogenization of the series of all geomagnetic field elements (X, Y, Z, H, D, I and F) to the level of epoch 2015. Since all

series except D have a gap of about 10 years long in the middle of the 20th century, splitting each of them into two, the homogenization to the level of 2015 was done only for the series available after 1951 (with D series homogenized for the whole time interval 1866-2015). The COI geomagnetic field elements are available via the following addresses: COI original data – doi.org/10.5281/zenodo.4122066 (Ribeiro et al, 2020); COI homogenized data – doi.org/10.5281/zenodo.4122289 (Morozova et al, 2020).

## 1 COI metadata

In previous work by Morozova et al. (2014), the series of monthly values of the declination of the geomagnetic field (D) measured at the Coimbra Magnetic Observatory (COI) between 1866 and 2006 were analyzed for the presence of artificial homogeneity breaks (HB). A number of HBs related to the changes of the observatory location and changes or repairs of the instruments were found and, when possible, corrected. Here we present the continuation of the previous work, extending the

homogeneity analysis to the monthly series of the horizontal (H) and vertical (Z) components, and the inclination (I) measured at COI between 1866 and 2006. Finally, in 2006 the set of old instruments were replaced with modern ones. Accordingly, we applied the same methods to the series of all elements for the 2006-2015 time interval to make them in line with current measurements done with instruments installed after 2006. To the time of the analysis, the most recent values in the series were available for December 2015. Since from that time no changes in the instruments and procedures took place,

the addition to the corrected series of measurements done after December 2015 will not affect their homogeneity.

The analysis is supported by metadata relative to the instruments used to measure H, Z and I components between 1866 and 2015. More details about the Coimbra Observatory history and metadata can be also found in Pais and Miranda (1995) and Morozova et al. (2014).

The first geomagnetic measurements at the COI Observatory were started in July 1866 at the *Cumeada* site (40º 12.4' N, 8º

25.4' W, 140 m a.s.l.). In January 1932 the observatory was relocated to a new site, *Alto da Baleia* (40º 13' N, 8º 25.3' W, 99 m a.s.l.), where it is still operating. The records have gaps that start between 1939 and 1942 (depending on the element) and end in October 1951. A number of instrument replacements and repairs, as well as changes of the measurements and calculation procedures took place between 1866 and 2015. Those related to the measurements of H, Z and I are described



below and summarized in Tables S1-S3 in the Supplement. The information on the D component can be found in Morozova

et al. (2014) and is also summarized in Table S4 in the Supplement.

Different combinations of three magnetic elements can be used to fully specify the geomagnetic field vector. The particular combination is determined by the kind of instruments being used. The combinations of HDZ (cylindrical components) and XYZ (Cartesian) are commonly recorded by the relative instruments (i.e. variographs or variometers), while the combinations HDZ, HDI and DIF (spherical) are the most easily measured by absolute instruments (Parkinson, 1983). Table

1 shows the combinations of geomagnetic elements measured by the absolute and relative instruments in the Coimbra observatory from 1866 to 2015.

Concerning the absolute measurements, three different combinations of magnetic elements were registered in different periods of the observatory history. In the first period, between 1864 and 1951, the absolute observations were made for H, D and I. The Gibson unifilar magnetometer used in the first absolute observations of H and D, was replaced in January 1878 by

the Elliott & Bros. unifilar magnetometer, which was kept running until 1948 and from 1951 to 1955. The magnetic inclination (I) started to be measured with an inclinometer of Barrow in June 1866, which was replaced by a John Dover inclinometer in September 1876. This was kept in use until October 1935 when it was replaced by a Sartorius Earth inductor. The absolute measurements of I were discontinued in 1939. Despite the observatory relocation in 1932 to *Alto da Baleia*, relatively far from the city centre and the tram lines, the quality of data not only has not improved but worsened significantly

during the first 20 years in *Alto da Baleia* (1932-1951). The oversimplification of observation routines and the non-negligible perturbations mainly related to the aging and drift of old absolute instruments that were kept in use, as well as the incorrect installation of the newly acquired Askania variographs, can be regarded as the main reasons for the low quality of the data. In the following period, between 1951/52 and 2006, the absolute measurement of I was replaced by the measurement of Z, so that the calculation of the baselines for the magnetic variations (H, D and Z) was now obtained

directly from the absolute measurements of these elements (please see Tab. 1 for information about the measured combination of elements at different time intervals).

After 1952 the absolute measurements of H and Z were obtained respectively with a quartz horizontal magnetometer (QHM) and a zero balance magnetometer (BMZ). The QHM and BMZ magnetometers are known to be semi-absolute instruments (i.e. instruments subject to drift) being then necessary to periodically assess the measurement quality by comparing them

with standard instruments. During period October 1951 – October 1952 two QHMs (nos. 190, 191, loaned to COI by the National Meteorological Service of Portugal) were used interchangeably. In 1952 the Coimbra observatory acquired two new QHM instruments (nos. 220, 221 with measurement certificates obtained in *Rude Skov*, Denmark). They were compared (during November and December of 1952) with the QHM no. 190 and no. 191, and the results were considered consistent (according to the published yearbook data for 1952). Then, in January 1953, QHM no. 220 was adopted as the main

observation instrument. Shortly after, in 1955, a new magnetometer QHM (no. 307) was acquired. This instrument was compared to QHM no. 220 (with consistent measurements) and was used as the main instrument for absolute measurements of the horizontal component from beginning of July 1955 until May 2006, while installed on pillar no 7. The absolute measurement of the Z component at the *Alto da Baleia* site started in December 1951/January 1952 with a BMZ no. 69, which was replaced in January 1953 by the BMZ no. 80. This was replaced in January 1977 by the BMZ no. 130 that assured

exclusively the absolute determination of the vertical component until May 2006.

The installation of the current set of modern instruments at the COI observatory began in 2006, with the DI-flux for the absolute observations (January 2006). This instrument is a combination of the fluxgate Mag-01H magnetometer (Bartington) with a universal theodolite YOM MG2kP. It replaced the classic declinometer Askania that had been in use since 1955 in the absolute determination of D. This new instrument was installed on a different pillar (no. 4, henceforward, the observatory's

reference pillar) in the absolute house, after the azimuth sighting has been determined for a new mark (cross of Missionary Sacred Heart located approximately at 750 m). All these changes resulted in a D baseline jump (Morozova et al., 2014).



On the other hand, until the end of May 2006, the baselines of components H and Z continued to be determined using the absolute values obtained by the QHM (no. 307, used since 1955) and BMZ (no. 130, used since 1969) magnetometers. These two instruments were replaced in the absolute observations in June 2006, and the baselines of H and Z started being

computed through the absolute observations of D, I and F, obtained with the DI-flux (D and I) and the proton magnetometer Geometrics G-856 (F). These equipment changes introduced discontinuities in the baselines of the H and Z series. In the case of H, there was a jump of -34 nT (considering the difference between values of June and May). In the case of component Z, there were two discontinuities: the first, also related to the change in the equipment of absolute measurements, resulted in a jump of -24 nT (considering the difference between the values of June and May), while the second, characterized by a jump

of -25 nT (considering the difference between the August and September values), was related to the replacement of the analog variometers (Eschenhagen model) by the digital (FGE model). In May 2007, the proton sensor, initially installed at a height of about 2 m near the entrance of the pavilion of absolute measurements, was moved to the top of pillar no. 5. Later, in July 2007, the old proton magnetometer G-856 was replaced by the new Overhauser GSM-90F1 magnetometer (GEM systems), installed in the variometer house. This last modification apparently did not introduce significant discontinuities in

the evolution of baselines. In addition, during most of the year 2014 it was not possible to carry out the absolute observations due to the failure of both the DI-flux (between February and September) and the proton magnetometer (between May and September). Both instruments were repaired by the respective companies (Bartington and GEM systems). During the period of interruption of the absolute observations the baselines calculated for January 2014 were used.

## 2 Homogenization procedure

The homogenization procedure used in this study is similar to the one described in detail in Morozova et al. (2014). The homogeneity study was done using both the visual analysis and a statistical homogeneity test, namely the standard normal homogeneity test (SNHT, see Alexandersson and Moberg (1997)), which allows the estimation of the statistical significance of homogeneity breaks (HB). SNHT is known to be oversensitive to HBs near the beginning and/or the end of the analyzed series (Costa and Soares, 2009). In general, the relative amplitude of a maximum of the SNHT statistics does not directly

depend on the strength of corresponding HB. The ratio of amplitudes of different maxima depends on the length of the studied period, relative distance between the breaks and/or to the proximity to the beginning/end of the series. Also, the correction of a break can result in changes of the relative amplitudes of other SNHT statistics' maxima. The natural cyclic variations and long term trends of the measured parameters are also detected by the statistical homogeneity tests and have to be accounted for.

Contrary to Morozova et al. (2014), we found that to take care of all significant artificial homogeneity breaks we have to analyse not only the time derivatives of the geomagnetic series but the series themselves. In some cases abrupt changes in a series mean level are clearly seen in the visual analysis of a geomagnetic element but not in the homogeneity test of its time derivative (e.g., the change of the mean level of the H component in March of 1922 discussed in Sec. 3.1), most probably shadowed by more significant inhomogeneities in the series.

The data series of geomagnetic elements and their time derivatives were analyzed using relative homogeneity tests applied to differences between COI series and similar data obtained from other geomagnetic observatories or simulated by a model which we refer to as "reference series". The series of these differences are denoted by adding the prefix "Δ". The series of time derivatives were calculated as differences between the same months of two consecutive years. Since this time derivative is close to the standard geomagnetic secular variation, we will denote it "SV". The original series of the analyzed

components are shown in Fig. 1 as red lines. The SV series are shown in the Supplement (Fig. S1, red lines).

Three types of series were used as references. The first type are observations from other European geomagnetic observatories (EO). The full list of these observatories is in the Supplement, Table S5. The values of geomagnetic elements





can be significantly different for different stations; however their SV series tend to be more similar. Therefore, the data from the reference observatories were used to calculate the average of first time derivative of the components ($SV_{EO}$) and the resulting series were then compared with $SV_{COI}$. The $\Delta SV_{EO}$ and $SV_{EO}$ series are shown in Figs. 2c-5c and Sup. Figs. S1, respectively, as green/yellow and green lines, respectively. The raw and SV individual Eos series were not used for the relative homogeneity tests because the data from individual observatories have many gaps and probably contain yet not corrected artificial homogeneity breaks which can, in turn, affect the homogeneity analysis of the COI series. However, they can be used to segregate homogeneity breaks of natural origin from artificial ones since the mean $SV_{EO}$ and $\Delta SV_{EO}$ series have no gaps due to the overlapping of the series from individual observatories.

The second type of reference series used to compare with COI data was built from simulated variations of the components calculated for the COI location using the COV-OBS model for the internal geomagnetic field (Gillet et al., 2013). As in the previous study (Morozova et al., 2014), two versions of the model were calculated: (1) using COI geomagnetic data among the set of all magnetic observatory ("all") and (2) excluding COI data from the internal field model calculation to mitigate its possible influence on the modelled series ("w/o COI"). The "w/o COI" model was computed by N. Gillet for the Morozova et al. (2014) study and used again in the present study for consistency. The comparison of the "all" and " w/o COI" predictions (see Sup. Fig. S2) shows that for the H component the differences (in the absolute values) between the two models do not exceed 2-2.5 nT during the 20[th] century but for more ancient epochs, in the 19th century, they grow up to 12 nT. Similar differences are seen for the I component: they do not exceed (in the absolute values) 0.3' during the 20th century but increase to 1.2' in the 19th century. Since the Z component was directly measured only during 2[nd] half of the 20[th] century, when we can find a significantly large number of geomagnetic observatories over Europe, the input data from the individual observatory (COI) does not significantly affect the model prediction, and the difference (in the absolute values) between the two models does not exceed 0.6 nT. We verify that even for the 19[th] century the results of the statistical homogeneity tests obtained using both "all" and " w/o COI" models are essentially the same, thus only the results for the "w/o COI" model are shown here denoted for simplicity as "COV-OBS". In our analysis in search for possible HBs we tested (1) the differences between the measured and simulated COI geomagnetic elements ("$\Delta$" series, see Figs. 2a-5a, red lines) and (2) the differences between the SV series of the measured and modelled components ("$\Delta SV_{COV-OBS}$" series, see Figs. 2c-5c, red/blue lines; see also Sup. Fig. S1 for SV of the COV-OBS series, black lines).

Differences between measurements and simulations arise not only from artificial inhomogeneities in the measured series but also from natural sources, e.g., solar and geomagnetic activity and crustal and induced fields that are not accounted for by the COV-OBS model. The H, I and Z series were found to be more strongly influenced by the solar activity than the D series analyzed in Morozova et al. (2014). In order to take this natural variability into account we used, as a third type of reference series, the following indices to describe the geomagnetic activity level (see Figs. 2b-5b): the index of inter-diurnal variability (IDV) from 1872 onward (Svalgaard et al., 2004; Svalgaard and Cliver, 2005), the aa index from 1868 onward, the global Kp index from 1951 onward (Menvielle et al., 2011) and the local $K_{COI}$ index from 1951 onward, already homogenized in Morozova et al. (2014). The IDV and aa indices are global geomagnetic indices with the longest available series. They were used to estimate geomagnetic activity variation for the epoch before 1951 when K index was introduced. The IDV, aa and K indices are calculated using different methodology and, consequently, their variations may represent different features of geomagnetic activity; moreover, the difference between the Kp and $K_{COI}$ indices reflect differences in the geomagnetic activity on the global and regional scales. Thus all these indices can be useful in detection of homogeneity breaks in the series of geomagnetic field components related to the natural sources. The geomagnetic indices were used only for visualization of the geomagnetic cycles of external origin and for estimations of the approximate dates of the geomagnetic activity's minima and maxima.





The corrections for the artificial HBs (δ values) were calculated using the information from the COI Observatory's

yearbooks and logbooks as well as by comparing the values of the measured components during some time intervals before and after a break in question (see details in Section 3).

Although the COV-OBS simulations cannot give the base level for COI, a more or less constant difference is expected between data series and COV-OBS simulations due to (1) uncertainty of the simulation although the uncertainty of COV-OBS model changes a lot in time, being larger for ancient epochs and (2) the crustal field. Assuming that the difference

between the observations and COV-OBS is constant in time (slightly varying due to the geomagnetic activity cycles, see Sec. 3) and for a lack of better choice of the baselines, we choose to use the COV-OBS model level as an approximation for the actual base level of COI geomagnetic field components. We also paid attention to the cycles of the solar and geomagnetic activity to not overcorrect the observational data. Thus, in some cases, when COI data varies smoothly around COV-OBS level and these variations were in agreement with variations of geomagnetic indices, no corrections were applied. Please see

detailed descriptions of specific cases in Section 3.

The HB corrections were applied backward in time, starting from the most recent break resulting in a corrected series being in line with the most recent measurements. This is a procedure usually applied for homogenization of the meteorological data (Morozova and Valente, 2012); however it may not always work for the COI series due to the long data gap between 1940s and 1951. Nevertheless, all HBs are described here in the direct chronological order to keep the usual narrative

timeline.

The quality of the applied correction was estimated by calculation of (1) the homogeneity tests statistics for the homogenized series and (2) the centred root mean square error parameter (CRMSE, see e.g. Taylor, 2001 and Venema et al., 2012). CRMSE was calculated using Eq. 1:

$$CRMSE^2 = \sigma_D{}^2 + \sigma_R{}^2 - 2 \cdot \sigma_D \cdot \sigma_R \cdot r \qquad (1)$$

where $\sigma_D$ and $\sigma_R$ are the standard deviations of the analyzed and reference series, respectively, and $r$ is the Pearson correlation coefficient between the analyzed and reference series. In case the corrected series is more homogeneous than the original one, the CRMSE values for the corrected series will be lower than for the original (Venema et al., 2012). To compare CRMSE obtained for different geomagnetic elements and their derivatives, the CRMSE values shown here are normalized: they are divided by the standard deviations of corresponding original series.

To summarize, the homogenization procedure consists of the following steps (see also Fig. 6):

Step 1. Preliminary analysis of the series, corrections of the typos and OCR errors; detection of outliers, calculation of the Δ, SV and ΔSV series

Step 2. Visual analysis of the data and statistical homogeneity tests;

Step 3. Detection and analysis of HBs:

1. Selection of HBs with statistical significance of at least 95%;

    2. Checking of the available metadata and logbooks;

    3. In case the homogeneity test detects HB with a significance <95% but a change is registered in the yearbooks for that time, this break can also be accepted for correction;

    4. In case HBs with statistical significance of at least 95% do not coincide with changes registered in the yearbooks but

coincide with HBs in the series of geomagnetic indices, these HBs are excluded from the correction as caused by the solar and geomagnetic activity variations;

Step 4. Correction of the selected breaks using the most appropriate method discussed in corresponding sections 3.1-3.3;

Step 5. Examination of the corrected series through the visual analysis, the statistical homogeneity tests and the CRMSE analysis.

The original and corrected series of the H, I and Z components, corresponding Δ, ΔSV and SV series are shown in Figs. 1-5, and Fig. S1. The results of homogeneity tests for the original and corrected series are shown in Sup. Figs. S3-S9.



## 3 Homogenization of the COI series (1866-2006)

### 3.1 H component

The measured COI H monthly means series (H) can be split into two parts separated by a gap of about 10 years long. The
first time interval, from June 1866 to December 1941, contains measurements at the *Cumeada* and *Alto da Baleia* sites (*C-AdB* period); the second time interval, from October 1951 to May 2006, is related solely to the *Alto da Baleia* site (*AdB* period) – see Figs. 1a, 2-3, Sup. Figs. S1a-S1b, S3-S5, Tabs. 1-2 and Sup. Tab. S1. The series were checked with yearbooks and logbooks for typos and OCR errors. Only absolute measurements are taken into account. Unfortunately, logbooks and yearbooks do not provide enough information about all changes in the measurement procedures and instruments repairs.
Some records are too laconic, e.g., "*some problems*" are mentioned in logbooks without any description of their nature. Also, the analysis of the logbooks showed that from time to time observers (on different reasons) did not follow the recommended routine (e.g., monthly means can be calculated using only one or two weeks of measurements). As a result, there are inhomogeneities that cannot be confirmed by metadata. On the other side, some events that could in principle generate inhomogeneities (e.g., installation of a new instrument) did not produce a visible and statistically significant HB. This is the
case of the installation of a new instrument in January 1878 (Unifilar of Elliott) for measuring H and D components. It is possible that during the installation process the new instrument was calibrated in line with the old one without the corresponding note added to the logbook. The aging of the instruments, an increase of the urban electromagnetic noise (the beginning of the electrical tram services in the nearby area) and other technical problems started to significantly affect the quality of the H data after 1929 (seen in Sup. Fig. S10a, pink area, as increased month-to-month variations). This situation
did not improve during the first years after the relocation in January 1932. This is clearly seen in the dispersion of the monthly H data (see Figs. 2a and 2c, and Sup. Fig. S10a). Only after the installation of the new instruments in 1951/1952 did the data quality improve (see Fig. 3a and 3c, and Sup. Fig. S10a). This irregular behaviour is also detected by the homogeneity tests statistics (Sup. Fig. S3). To reduce the influence of this odd time period, homogeneity tests were calculated for two time intervals: from 1866 to 1941 (whole *C-AdB* period) and from 1866 to 1933 (a period excluding the
epoch of major irregularities) – see Sup. Figs. S4.

The analysis of the first part of the H series showed that there were two statistically significant HBs (see Table 2): around March-April 1922 and from July to October 1931. In our opinion, the main reason for the break in March-April 1922 is the construction of a building within about 50 m from the house of absolute instruments. Despite the fact that there is no clear confirmation of this date in the Observatory logbooks and yearbooks, it was chosen for the corrections both because it is
clearly seen in the visual analysis and due to its high statistical significance (> 99%, Sup. Figs. S3-S4). The latter HB or, more precisely, a four months long jump of the mean level in July – October 1931 (Fig. 2) was likely caused by a lack of regular measurements during this time interval which resulted in an artificially high level of the average monthly values. This conclusion is based on the analysis of the records and notes in the Observatory's logbooks. Also, this break nearly corresponds to the relocation period, and it is likely that the routines had changed with consequences in the data quality.
Therefore, this HB was also chosen for correction.

These two HBs are not the only features that can be detected in the analysis of the H series during the *C-AdB* period. First of all, we have to mention a long-term wave-like variation with minima around 1892 and 1932 and maxima around 1873-1878 and 1912 clearly seen in ΔH and SV (Fig. 2a and Sup. Fig. S1a, red lines). This variation is detected by the homogeneity tests applied to ΔH with a maximum of the SNHT statistics in 1880s (Sup. Fig. S3-S4). This variation is not seen in the ΔSV
series (see Fig. 2c and corresponding homogeneity test results in Sup. S3b-S4b). We compared ΔH COI series with similar Δ series (differences between the observations and the COV-OBS model predictions) from other observatories available for this time interval (see Table S5). For the period 1866-1940 there are data from eight observatories: Parc Saint-Maur (PSM), Perpignan (PER), San Fernando (SFS), Oslo (OSL), Prague (PRA), Greenwich (GRW), Munich (MNH) and Lisbon (LIS);



however the data sets of the MNH and LIS observatories have many gaps and contain clear shifts of the baseline (probably

caused by changes in the observatory location, instruments or measurements procedure). The Δ series for the H component measured at the remaining six stations are shown in Sup. Fig. S11. The data observed at the observatories closest to COI observatories, SFS and PER, also show similar wave-like variations; similar variations are probably seen in the data of PRA but since PRA data have significant gap and were, probably, not homogenized, we cannot draw a final conclusion. On the other hand, the Δ series obtained from the PSM and OSL data show not a wave-like variation but rather a steadily increase.

Finally, the trend of the Δ series obtained from the GRW data is ambiguous.

Therefore, taking into account the absence of the recorded changes in the station environment and the similarity between the time variations of the series measured by COI and closest observatories, we conclude that the observed long-term variations are due to the natural evolution of the geomagnetic field unaccounted for by the COV-OBS model.

The ΔH series also shows quasi-periodic decadal variations. These variations can also be found in the SV series, but there

they are blurred by the high variability of the data. The origin of these quasi-decadal variations is clearly in the changes of the solar (and, consequently, geomagnetic) activity level. Figure 2 shows variations of the ΔH and $\Delta SV_{COV-OBS}$ series (panels a and c, respectively, red lines) alongside with the variations of the aa and IDV indices (panel b). As one can see, the highest geomagnetic activity corresponds to lower values of ΔH. On contrary, the periods of low geomagnetic activity correlate with epochs of higher ΔH values. Although this variability has natural origin, from the statistical point of view it results in

inhomogeneities detected by homogeneity tests (see Sup. Figs. S3-S4). Nevertheless, the geomagnetic indices variations confirm the natural origin of these quasi-decadal variations of H and that they must be kept in the data.

Thus, only two HB found during the *C-AdB* period were selected for correction (in 1922 and 1931, see Tab. 2). The correction values (δH, Tab. 2) were calculated as the difference between the means of the original ΔH series calculated for a certain time interval before and after the date of HB. For the first HB this time interval was chosen to be 12 months. For the

second HB the length of this time interval was only 4 months – the time between the break (July 1931) and the end of the series (October 1931). The corrections were rounded to the nearest whole number (since the presision of the 19[th] century instruments was not higher than 1 nT). The corrected H, ΔH, SV and $\Delta SV_{COV-OBS}$ series are shown in Figs.1a, 2a and Sup. Figs. S1a as blue lines and the homogeneity test results for the corrected series are also shown in Sup. Figs. S3-S4.

According to the COI annual books and logbooks (Tab. 2, Sup. Tab. S1), the old instrument (Unifilar of Elliot) was kept in

use for the first three months of the *AdB* period. Then, in January 1952, a set of new instruments (QHM nos. 190, 191) was installed. The difference between the old and new instruments was considered "*insignificant*". Later, in November 1952, these magnetometers were compared to newly purchased magnetometers QHM nos. 220, 221. According to the logbooks, the difference between the new and old instruments was about 20 nT. The QHM no. 220 magnetometer started being used as the main instrument. This information is supported by the visual analysis of the ΔH series (Fig. 3a): there is a visible shift of

the main level of the ΔH series not related to the geomagnetic activity variations (represented by the geomagnetic aa, IDV, Kp and $K_{COI}$ indices in Fig. 3b).

Later, in September 1953, the comparison between the measurements by the new QHM no. 220 and the old QHM no. 190 instruments showed that the new H values were higher by about 5.4 nT. This change of the mean level is also seen in the visual analysis. The other comparison of the COI and reference instruments took place in August 1959. Both the visual

analysis and homogeneity tests show no significant changes in the data main level at this date. We assume that for this time period the difference was too low to be seen comparing to the level of the monthly data variability.

One more comparison between the COI and reference instruments took place in April 1968. The differences were found to be about 18 nT. However, the analysis of the ΔH series shows that this correction must be applied up to April 1969. The last recorded comparison of the COI and reference instruments in August 1976 seems to cause no shifts of the H series that can

be detected by the visual analysis or homogeneity tests.





There are also two outliers, February 1982 and October-December 2001, respectively, that have no direct documental support; however, as mentioned in the logbooks, in October of 2001 the absolute measurements were taken only during the third decade of the month, and we consider this as a possible source for the latter outlier.

The analysis of the homogeneity tests statistics (see Sup. Fig. S5) shows that the ΔH and $\Delta SV_{COV-OBS}$ series contain both the

artificial inhomogeneities (above mentioned shifts and outliers) and the natural periodicities related (mostly) to the solar and geomagnetic cycles. Taking into account the results of the visual analysis, homogeneity tests and logbook records, we selected only five HBs for the corrections, listed in Tab. 2. The corrections for HBs associated with comparison to the reference instruments mentioned above were taken from the logbooks and the corrections for the outliers were calculated as differences between the means of the original ΔH series 4 months before and after each of the outlier in 1982 and before

beginning and after end of the outlier interval in 2001 (see values in Tab. 2). Such short time interval was chosen to avoid contamination from the cyclic variations of the geomagnetic activity – compare Fig. 2a and 2b. The corrections were rounded to the nearest whole number (the precision of the instruments was 1 nT). The corrected series are shown in Figs. 3 and Sup. S1a as blue lines and the homogeneity test results for the corrected series are also shown in Sup. Figs. S5.

In some cases the corrections of the artificial HBs resulted in an increase of the statistical significance of the "natural HB"

related to the geomagnetic cycles (Sup. Figs. S5a). The fact that this behaviour is seen mostly during the *AdB* period can be due to the higher precision and precision of the new instruments; another possible reason is the increase of the geomagnetic activity in the 2[nd] half of the 20[th] century reported by, e.g. Love (2011). Based on the variations of the geomagnetic indices, we conclude that the maxima observed on the SNHT statistics around 1980s, 1990s and 2000s (seen in Sup. Fig. S5a) are due to the natural variability of the data and do not need correction.

The CRMSE values for the original and corrected H and SV series relative to the corresponding reference series are shown in Fig. 7a. The results of this analysis support the conclusion that the corrected series are more homogeneous than the original ones; however, they contain "natural" inhomogeneities related, in particular, to the variations of the solar/geomagnetic activity.

### 3.2 I component

The digital series of the I monthly absolute measurements available from different sources (including the World Data Centre for Geomagnetism, Edinburgh, UK, website) starts in June 1866 and ends in May 1940. The series contains a number of gaps (from 1 to 6 months long, with longer gaps clustered between 1936 and 1940). This series was checked for the OCR errors, typos and calculation errors. The homogeneity of the series was affected by the relocation of the Observatory (in 1932), repairs and replacement of instruments or its parts, and environmental changes (urban electromagnetic noise level).

The I series and its derivatives are shown in Figs. 1b, 4 and Sup. Fig. S1b. As one can see, between 1866 and 1932 the Δ and ΔSV series (Figs. 4a and 4c) show fluctuations related both to the already known changes in the instruments/procedures and to the variations of the solar activity (as is clearly seen in Fig. 4b where the IDV and aa geomagnetic indices are plotted): the Δ series varies in phase with IDV and aa. However, starting from about 1931 the I series shows a strong and unexpected decrease that cannot be explained neither by natural variations of the field nor by events mentioned in the metadata. The

most likely explanation is the degradation or incorrect installation of the instruments/needles or other (human) factors (e.g., disregard of the measurement procedures). Besides, the records in the logbooks clearly state that the I component was measured at COI only until the end of 1938. The origin of the (highly fragmented) monthly values of I between January 1939 and May 1940 is unknown. Therefore, we strongly recommend to not use the COI I series for this period and, accordingly, the I values for this time interval are removed from the final homogenized version described in this work.

The COI I series containing the measurements between 1866 and 1938 was submitted to the homogenization procedure. The homogeneity tests statistics were calculated both for the whole interval (1866-1938) and for a shorter time interval (1866-1930), shown in Sup. Fig. S6-S7, respectively, to exclude the period of the fast decrease. Both the visual analysis (Fig. 4)



and the statistical homogeneity tests (Sup. Fig. S6-S7) show a number of HBs that are related to the known changes of the instruments or station location (see Tab. 2 and Sup. Tab. S2). These dates are: September 1876 (installation of the inclinometer of Dover), November 1922 and September 1928 (problems with inclinometer's needles, construction of a house near the geomagnetic absolute house and the installation of a spectroheliograph there), January 1931 when measurements were simultaneously taken at two locations (*Cumeada* and *AdB* site), January 1932 (official re-location of the Observatory), January 1939 (beginning of unexplained decrease of the I values). These events are listed in Table 2 (and Sup. Table S2). There is also a HB in May 1883 that is seen in the homogeneity tests of the $\Delta SV_{COV-OBS}$ series (Sup. Fig. S7b); however the yearbooks and logbooks of the Observatory contain no information that can be related to this HB. The comparison of the $\Delta$ and the geomagnetic indices series (Fig. 4) shows that this HB is of natural origin and is related to a sharp decrease in the geomagnetic activity. Thus, this HB was not corrected. It has to be mentioned that the analysis of the data for the time interval 1931-1932 lead us to assume that although the logbook records mention that during 1931 the measurements were taken at the *Cumeada* and the *AdB* sites simultaneously, the published measurements, most likely, were obtained at the new site. Furthermore, the change of the instrument in October 1935 (the old inclinometer was replaced by an earth inductor) during the period of the fast decrease of I though seen both in the visual analysis and homogeneity tests, cannot be corrected due to the overall bad quality of the data.

As shown in Fig. 4a, during the time interval from November 1922 to August 1928 the $\Delta$ series slowly fluctuates (due to geomagnetic activity cycles) around zero, meaning that the differences between the measured and simulated I values do not have any systematic shift or trend. Assuming that there is no reason for a sudden (on the time scale of 1 month) change of the base level of the order of 5-10', we decided to use COV-OBS model as an approximation for the actual COI I base level. Therefore, the time interval from November 1922 to August 1928 was chosen as a reference interval: the corrections for the I series were calculated in a way that the mean values of $\Delta$ for any interval between HBs were equal to zero (the average value of $\Delta$ for the homogeneous interval from November 1922 to August 1928). Three time intervals between HBs were corrected (Tab. 2): June 1866 – August 1876, September 1876 – October 1922 and September 1928 – December 1930. The corrections were rounded to 1' (according to the precision of the instrument). The time interval between January 1931 and December 1938 was not corrected due to the bad data quality and strong decreasing trend. The corrected I series and its derivatives are shown in Figs. 4 and Sup. Fig. S1b (blue lines) and corresponding homogeneity tests statistics are in Sup. Fig. S6-S7. The homogeneity tests statistics for the corrected I series still show statistically significant HBs related to geomagnetic activity cycles. The CRMSE calculated for the original and the corrected series are shown in Fig. 7b. The CRMSEs for the corrected series are lower than for the original. Thus, the corrected I series is more homogeneous than the original one and contains, mostly, variations related to the natural sources, except the period from 1931 to 1938 which we do not recommend to use for any scientific analysis or simulation.

### 3.3 Z component

The installation of a new BMZ instrument in January 1953 is seen in the Z records (Figs. 1c and 5, and Sup. Fig. S1c) as a sudden step-like change of the baseline by about 30 nT. The visual analysis of the Z and $\Delta Z$ series (Figs. 1c and 5) show that after about 1.5 year from that date some problems with the instrument or measurement procedures started to appear: the level of the Z series jumped up by about 20 nT in the middle of 1954 and again by about 10nT in the end of 1955. Yet another increase of the base level for Z series took place in the end of 1960. The reasons for these changes are unknown since the currently available logbooks do not contain any notes on the Z measurements until 1963. However, these jumps cannot be neglected since they are seen not only in the visual analysis but also in the relative homogeneity tests of both the $\Delta$ and $\Delta SV_{COV-OBS}$ series (see Sup. Fig. S8-S9). These HBs do not coincide with the cycles of the geomagnetic activity (see geomagnetic indices aa, IDV, Kp and $K_{COI}$ in Fig. 5b) and they are too sudden (see also Sup. Fig. S10c) to be attributed to the solar activity effect. They are also not seen in the variations of the $SV_{EO}$ series (Sup. Fig. S1d). The BMZ no. 80 was





replaced by the BMZ no. 130 in January 1977. Plus, two HBs were detected around September 1971 and December 1973. It seems that some measures were taken to improve the quality of the Z measurements at COI: from December 1973 the base line of the recorded measurements is in a good agreement with simulation done by the COV-OBS models. This homogeneous period started to deteriorate around 1982 when an annual cycle appeared in the data. This cycle became more prominent around 1989. Besides, the decrease of the mean Δ in 1990-1991 was too steep comparing to the variations of the

geomagnetic indices (Fig. 5b). Similar to the I component, we decided to use COV-OBS model as an approximation for the actual COI Z base level assuming that there is no natural sources for a sudden (on the time scale of 1 month) changes of the base level of the order of 5-20 nT.

The origin of the annual cycle (and, probably, a part of the overall decrease of the Z base level between 1990 and 2006) seems to be related to the degradation of the instruments (the precision of the instruments to the temperature variations

strongly increased), or, perhaps, errors in the calculation due to non-application of the temperature correction factor (or application of a wrong factor). This assumption is supported by the comparison of the monthly mean ΔZ values with the monthly mean temperature parameters measured by the COI meteorological observatory (see Morozova and Valente, 2012) – daily minimum (Tmin) and maximum (Tmax) temperatures, see Sup. Fig. S12. For the time interval 1990-2006 the correlation coefficients between the ΔZ and meteorological series are about 0.7 ($p \leq 0.01$, calculated using a Monte-Carlo

approach with artificial series constructed using a bootstrapping with moving blocks procedure, block length equals 12 months).

After the visual and statistical analysis of the Z, Δ and ΔSV series, several HB classified as "artificial" were identified and selected for correction (please see also Fig.S8-S9). They are listed in Table 2 (see also Sup. Table S3).

The time interval June 1954 – October 1960 was divided into two: before and after October 1955. This additional HB seen

only in the visual analysis was needed to avoid step-like variations in the corrected Z series. This HB seems to be too small to be seen in the homogeneity tests on the background of the signal from other breaks. However, it becomes visible and statistically significant if the corrections are applied to all other HBs. Similar to the I component case, taking the value ΔZ = 0 nT (the mean ΔZ value for the time period 1973 December – 1989 October) as the reference level we calculated the correction value for this HB (see also Fig. 5).

The annual cycle (from 1990 to 2006) was not corrected because linear regression coefficients of Z series on the temperature series change from year to year. It is possible that such corrections for the meteorological effect are nonlinear, should take the average humidity level into account, and, also, depend on the mean annual level of the geomagnetic activity.

The corrected Z series and its derivatives are shown in Fig. 5 and Sup. Fig. S1c as blue lines. The homogeneity tests (Sup. Fig. S8-9) show that the corrected series contain HBs coinciding with the geomagnetic activity cycles (e.g., the maximum in

1980s) and the annual temperature cycle in 1990-2006. The CRMSE values for the original and corrected Z and SV series relative to the corresponding reference series are shown in Fig. 7c. The CRMSE of the corrected Z series are lower than those of the original ones both when the whole series (1951-2006) and a shorter time interval (1951-1988) are considered. The results of this analysis show that the corrected series are more homogeneous than the original ones; however, they still contains "natural" inhomogeneities related to the variations of the solar/geomagnetic activity and the meteorological effect

for the 1990-2006 time interval.

**4 Homogenization of the historical COI series in line with the current digital measurements (2006-2015)**

To complete the procedure of the homogenization of the historical COI geomagnetic series, the series for all geomagnetic elements (X, Y, Z, H, D, I, and F) were obtained for the period 1866-2006 using the direct measurements for some elements and calculating the other elements from the measurements as described below. The series of the geomagnetic elements were

homogenized in line with most recent values in the series used for this study, i.e., December 2015, and corrections are shown





in Table 2. The non-corrected and corrected series for the time interval 2006-2015 are shown in Fig. 8 together with the COV-OBS predictions.

First of all, we compared measured series before and after 2006 to make necessary corrections. To keep D series in line with current measurements we introduced additional correction to the series previously homogenized in Morozova et al., 2014.
The correction for the D series of -7.3' (considering the difference between the monthly values of December 2005 and January 2006) was applied to the series between 1866 and 2005 (Tab. 2). The correction for the I component before June 2006 is not necessary since there is no visible jump. The corrections for the F series were calculated backward in time, first for the data before May 2007 (the change of the instrument implied a jump in the baseline between the months of April and May 2007) and later for the data before June 2006 (Tab. 2). The corrections were calculated assuming that the average
change of the F values on the monthly scale is about +10 nT/yr or +0.83 nT/month, which is the average rate of change of F between June 2007 and December 2015. The corrections for F were applied only for the 1951-2005 time period. From the corrected D and F and from the uncorrected I series the new series of the H and Z for the 2006-2015 time interval were re-calculated. As a result, no visible breaks were seen in the H series, whereas there seems to be a significant jump (estimated as about 59 nT) in the Z series. This correction (Table 2) was applied to the whole period 1951-2005. Please note that this
correction also solved the problem of the voluntary chosen baseline level when the Z series for the 1951-2006 time interval was corrected (see Sec. 3.3).

Afterwards, all the elements (except D) were calculated for those periods were they were not measured:
– Z: May 1864 – December 1938 (from the H and I data);
– F: May 1864 – December 1938 (from H and I) and October 1951 – May 2006 (from H and Z);
– I: October 1951 – May 2006 (from H and Z).

Also, the X and Y components were calculated from July 1867 (beginning of the D observations) to December 2015. All the series except the D series have a gap in the data (from 1938/1941 to 1951). The series before the gap were not homogenized in line with the current observations both because of the gap and due to the bad quality of the data at the end of the *C-AdB* period. As was mentioned above, in most cases, the COV-OBS predictions were used as a reference level for the 1866-1951
time interval. All the series of the COI geomagnetic field elements (original and corrected) are shown in Fig. 1 (red and blue lines, respectively).

## 5 Conclusions

As a continuation of the previous work (Morozova et al., 2014) on the homogenization of the historical series of the Coimbra Geomagnetic Observatory (COI), Portugal, we applied here similar analysis to the long time series of H (horizontal component),
component), Z (vertical component) and I (inclination) absolute monthly means. The H component was measured from 1866 to 2006 with a break of about 10 years in the middle of 20[th] century, the I component was measured from 1866 to 1938 and the Z records are from 1951 to 2006. Monthly mean values of the analyzed parameters and their time derivatives were studied relative to the data series from both other European observatories and prediction made for the Coimbra location by the COV-OBS model of the geomagnetic field. The analysis was done using available metadata (logbooks and annual books
of the Observatory), the visual analysis of the series and the SNHT statistical homogeneity test. Changes of the geomagnetic activity were accounted for by the comparison to the geomagnetic activity indices (aa, IDV, Kp and the local $K_{COI}$ index). Statistically significant homogeneity breaks of artificial (non-natural) origin were detected and whenever possible corrected. The corrected series were tested for homogeneity.

Overall, the corrected series show more consistency with the data from other geomagnetic observatories. The series recorded
after 1951 (the Z and the 2[nd] part of the H series) have better data quality compared to the series recorded during the 19[th] and the 1[st] half of the 20[th] centuries, allowing a much easier detection and correction of the homogeneity breaks. This, mostly,



results from a better quality of the instruments (lower measurements errors) and improved observation procedures. Unfortunately, the corrected series still contain artificial inhomogeneities related, in particular, to the degradation of the instruments and an increase of the urban electromagnetic noise.

The final step of the homogenization of the COI series consisted in the homogenization of all the series to the level of the current measurements. The series of the F, D and I components available before 2006 were homogenized in line with the current observations. Then, the series of other elements were re-calculated for the period 1866-2006 and, when needed, corrected for jumps. Please note that all series, with the exception of D series, available from 1866 to 1951 are not homogenized in line with current COI measurements due to the about 10 year long break in the measurements.

Currently, the original and homogenized series at **doi.org/10.5281/zenodo.4122066** and **doi.org/10.5281/zenodo.4122289**, respectively, and can be used in studies of secular variation. Besides, the metadata for the COI historical geomagnetic series are also summarized in Sup. Tab. S1-S4. During the second half of the 19th and first half of the 20th centuries, historical observatories represent the main source of highly dynamic and relatively precise geomagnetic data. We believe that applying this kind of study to other historical series can contribute to better constrain geomagnetic field models as COV-OBS, and 470 also to better characterize cycles of external geomagnetic activity during the pre-satellite era.

**Author contribution**

PR provided digital series of the COI geomagnetic field elements, worked with the logbooks and annual books. AP provided COV-OBS simulations. AM performed visual and statistical analyses of the series and estimated corrections. AM prepared 475 the manuscript ; PR and AP performed revision of the manuscript.

**Acknowledgement**

CITEUC is funded by National Funds through FCT - Foundation for Science and Technology (project: UID/MULTI/00611/2019) and FEDER – European Regional Development Fund through COMPETE 2020 – Operational Programme Competitiveness and Internationalization (project: POCI-01-0145-FEDER-006922).

A. Morozova thanks Geophysical and Astronomical Observatory of the University of Coimbra and Dr. J. Fernandes for financial support of this work.

This study is also a contribution of the project HISTIGUC (PTDC.FER-HFC.30666) and the project MAG-GIC (PTDC/CTA-GEO/31744/2017).

A. Pais wishes to thank Nicolas Gillet for kindly computing the "COV-OBS w/o COI" model that was used in this study..


**Data availability**

The COI geomagnetic field elements are available via the following addresses: COI original data – doi.org/10.5281/zenodo.4122066 (Ribeiro et al, 2020); COI homogenized data – doi.org/10.5281/zenodo.4122289 (Morozova et al, 2020).

The historical temperature series of the Coimbra Observatory are available at Morozova, AL; Valente, MA (2012): Homogenization of Portuguese long-term temperature data series: Lisbon, Coimbra and Porto. doi:10.1594/PANGAEA.785377.

The IDV index can be downloaded from the Leif Svalgaard website, http://www.leif.org/research/.

The aa index is available from the NGDC website http://www.ngdc.noaa.gov/stp/spaceweather.html.





The global Kp index can be downloaded from http://www-app3.gfz-potsdam.de/kp_index/index.html or http://isgi.cetp.ipsl.fr/des_kp_ind.html.

**Competing interests**

The authors declare that they have no conflict of interest.

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



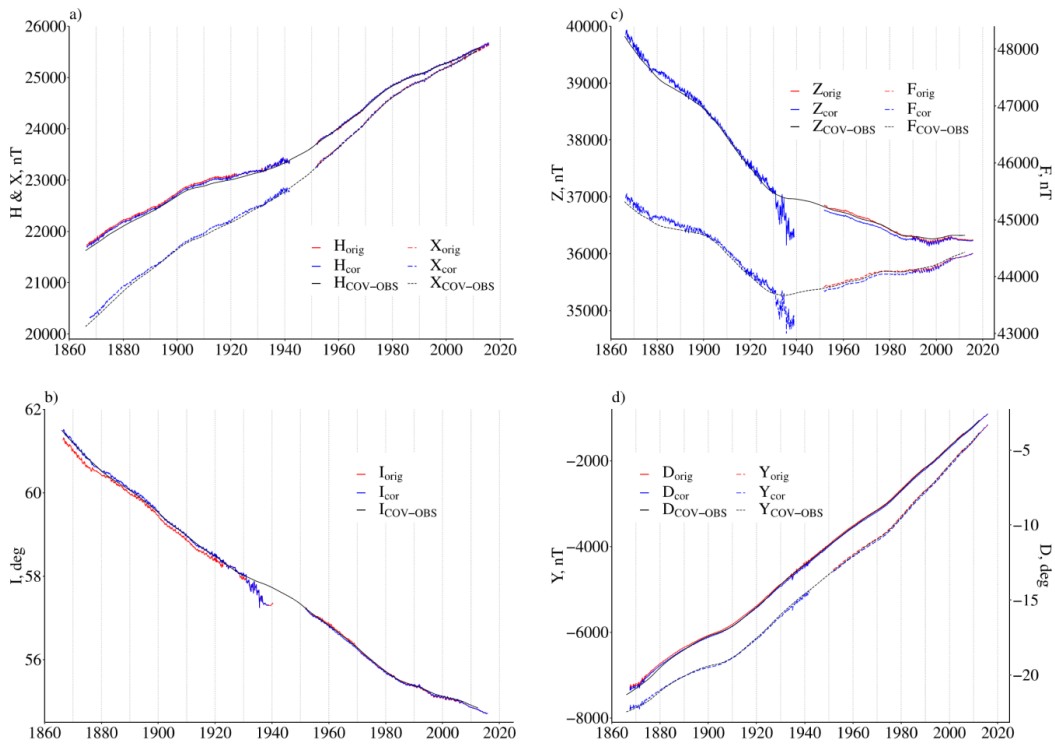


**Figure 1: For the 1866 to 2015 time period, measured (red) and corrected (blue) series of the COI geomagnetic field elements (a) H and X; (b) I; (c) F and Z; (d) D and Y. The COV-OBS model predictions are in black.**

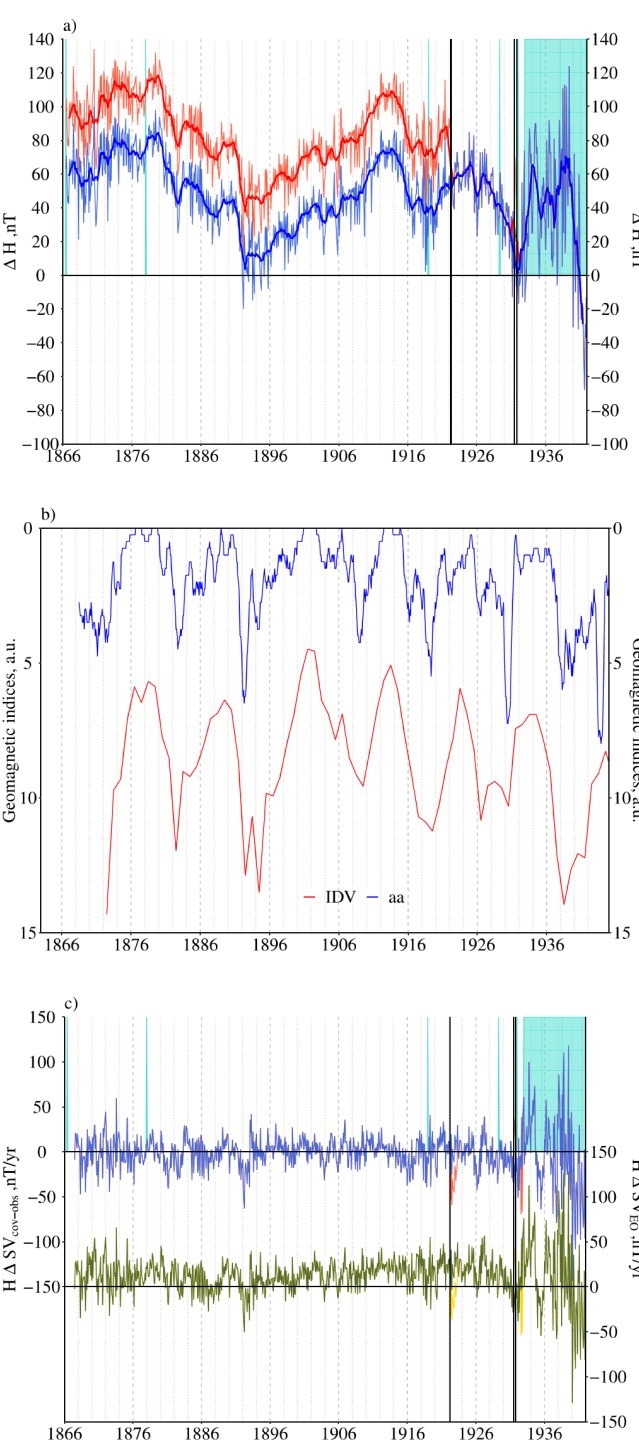

**Figure 2: (a) Original (red) and corrected (blue) COI ΔH series for the C-AdB period of measurements: 1866-1941. Thin lines are**
**data and thick lines are moving average with 12-month span window. (b) Variations of the geomagnetic indices aa (monthly data, blue) and IDV (annual data, red). Please note reversed Y axis. (c) Original (red) and corrected (blue) ΔSV$_{COV-OBS}$ series for H; original (yellow) and corrected (green) ΔSV$_{EO}$ series for H. Cyan vertical lines/rectangles mark possible HBs which are not corrected (see details in the text), black vertical lines mark dates of the corrected HBs.**



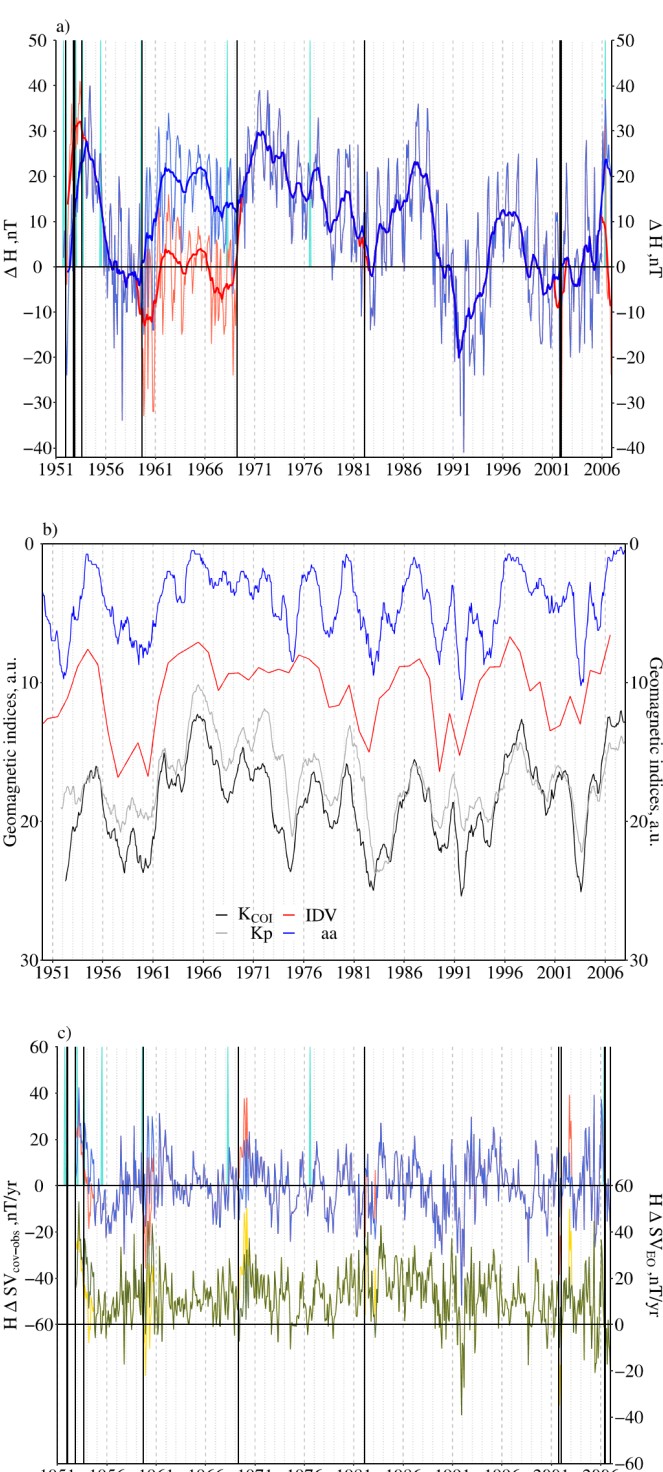

**Figure 3: Same as Figure 2 but for the *AdB* period of measurements: 1951-2006. On (b) the monthly sums of the local $K_{COI}$ (grey) and the global Kp (black) indices are also shown.**

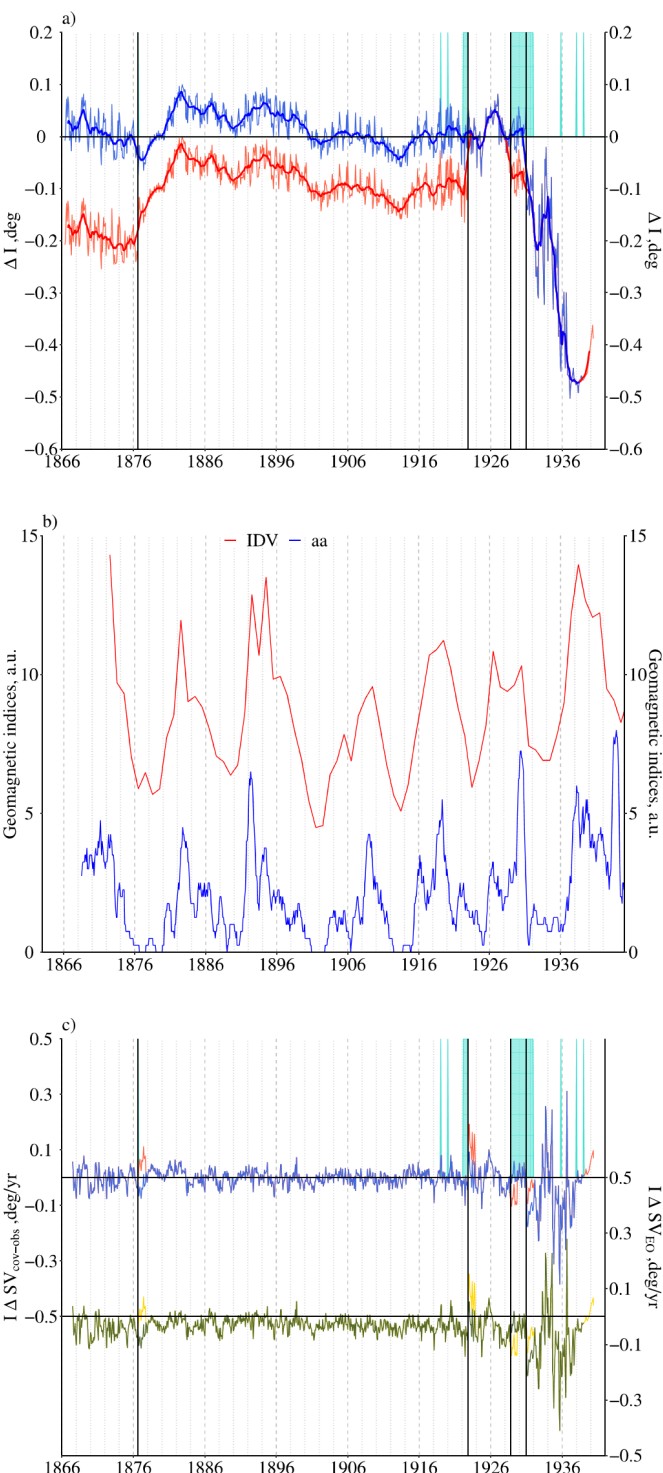

**Figure 4: Same as Figure 2 but for the COI I series.**

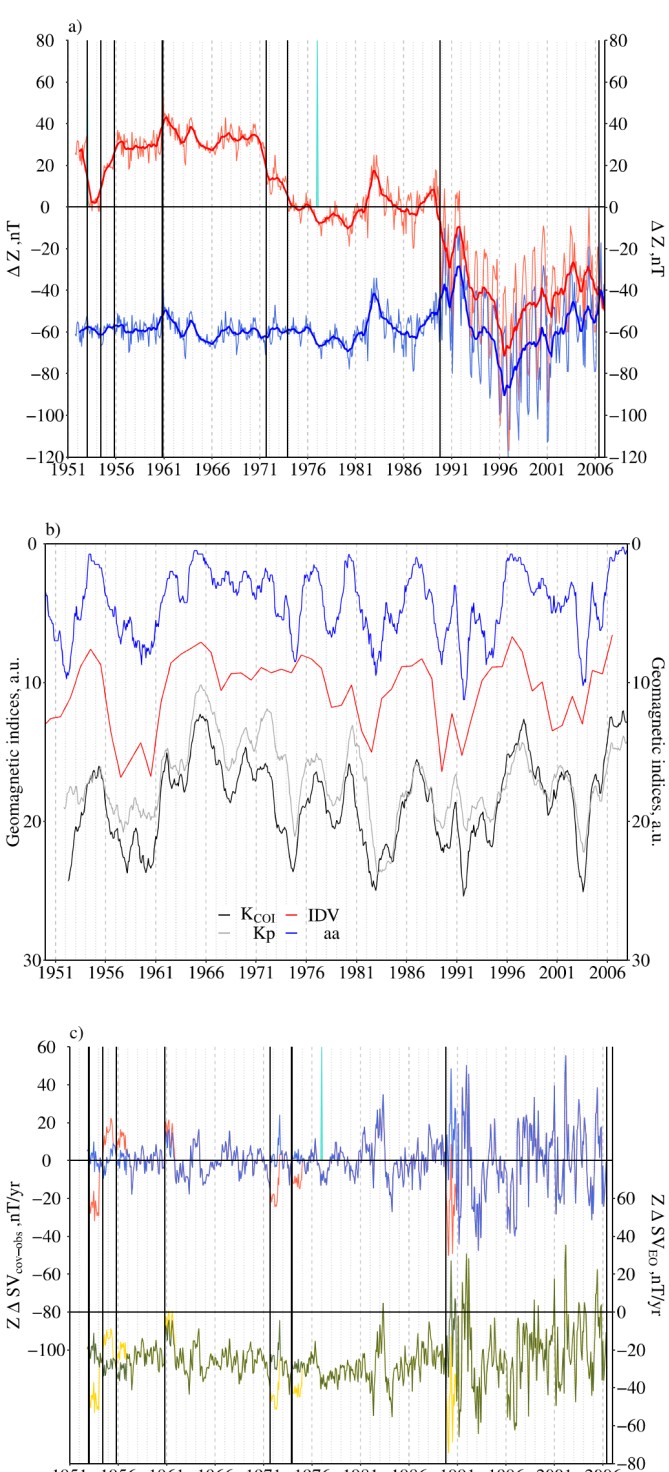

**Figure 5: Same as Figure 3 but for the COI Z series.**





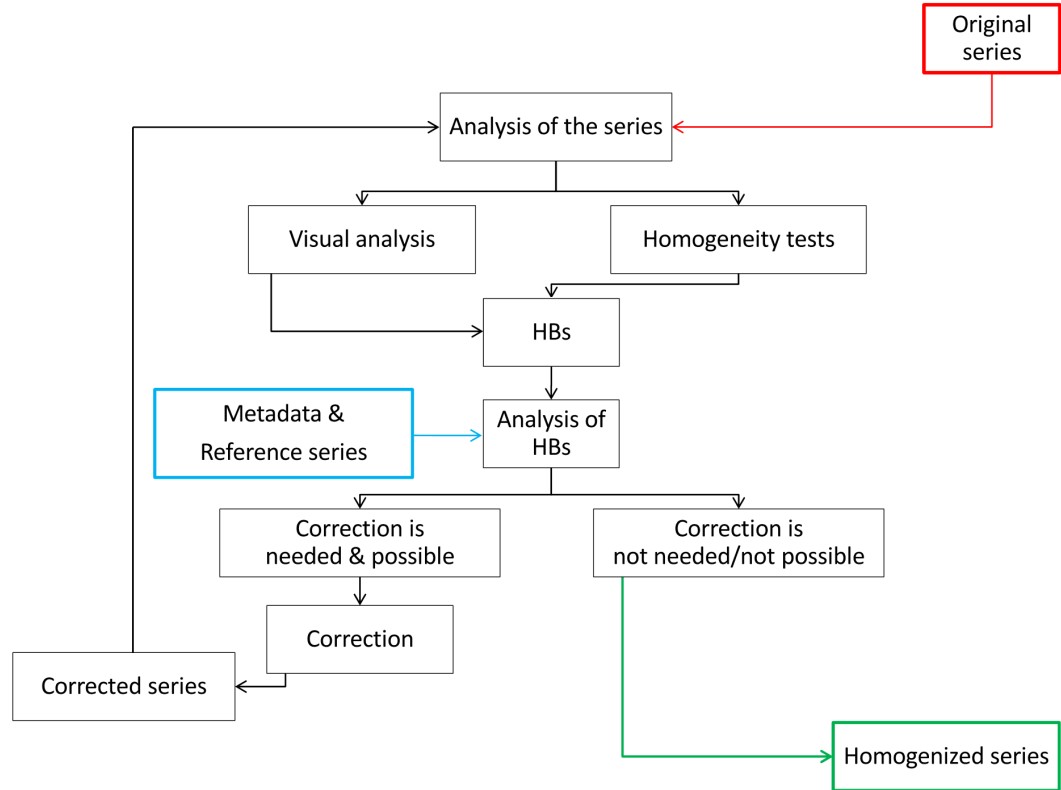

**Figure 6: Homogenization scheme**

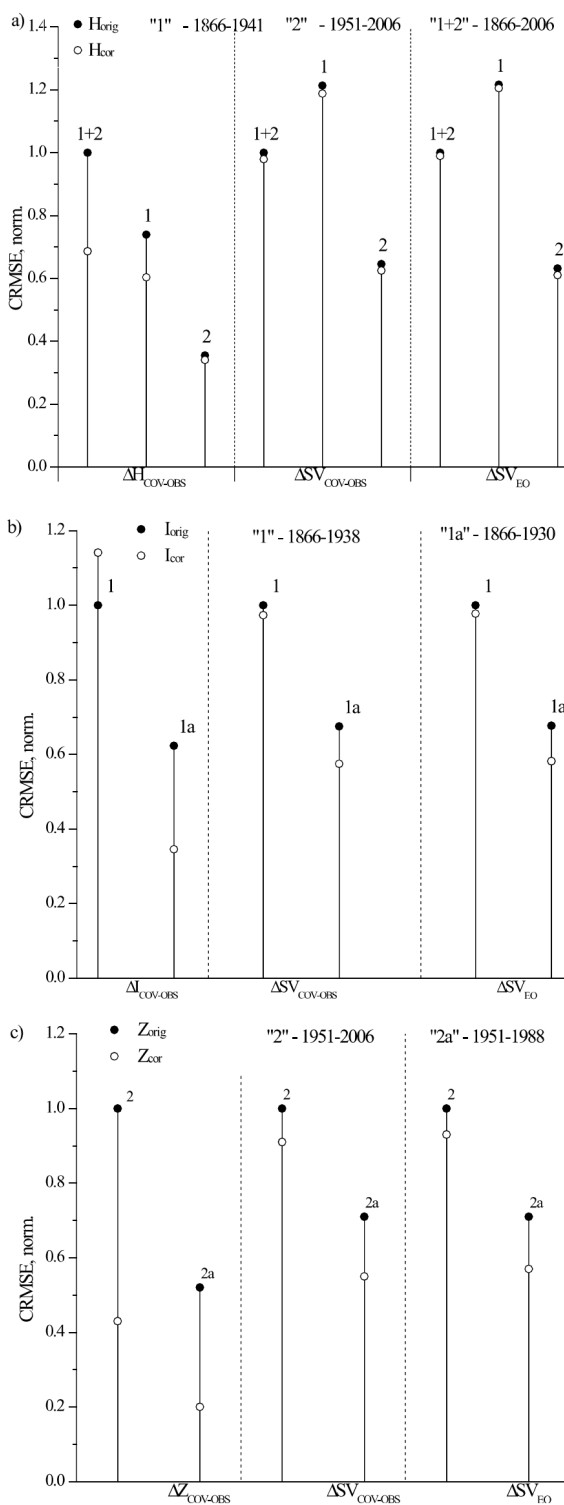

**Figure 7. CRMSE of the original (black dots) and corrected (white filled circles) COI series estimated for different time intervals: a - COI H, b - COI I, c - COI Z. CRMSE values are normalized by the standard deviations of the original series.**

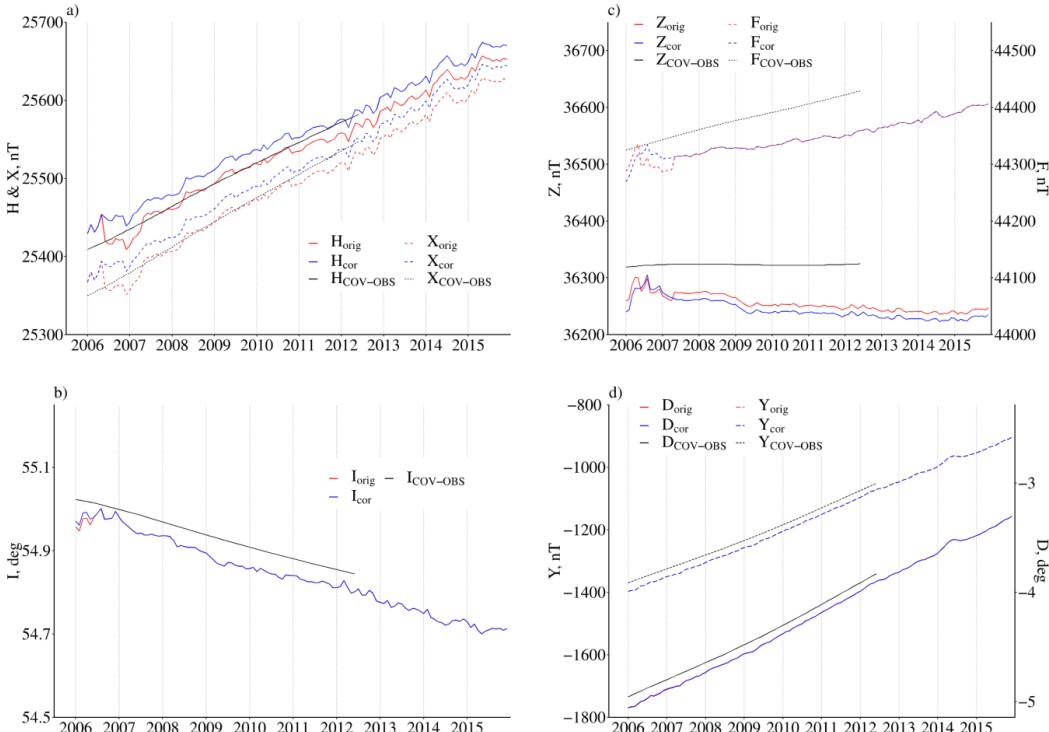

**Figure 8. Same as Figure 1 but from 2006 to 2015.**



**Table 1. Periods of the absolute measurements of the COI geomagnetic field elements (between 1866 and 2015)**

| Component | *Cumeada* site | *Alto da Baleia* site |
|---|---|---|
| D | 1866-1932 | 1932-1951<br>1951-2015 |
| H | 1866-1932 | 1932-1941<br>1951-2006 |
| I | 1866-1932 | 1932-1940<br>2007-2015 |
| Z | – | 1951-2006 |
| F | – | 2007-2015 |

| Measured elements | | | | |
|---|---|---|---|---|
| | 1864-1931 | 1932-1941 | 1951-2006 | 2007-2015 |
| Absolute measurements (baselines) | HDI | HDI | HDZ | DIF |
| Relative measurements (magnetograms) | HDZ | HDZ | HDZ | HDZ |




**Table 2. Metadata and proposed corrections (δ) for COI series for different time intervals. *C-AdB – Cumeada and Alto da Baleia sites*, *AdB - Alto da Baleia site*. "d.n.u." – it is not recommended to use the data for this time interval.**

| period | Time interval | Notes | δ |
|---|---|---|---|
| | | H (1866-2015) | |
| *C-AdB* | 1866 June–1922 March | calculation procedure change | -34 nT |
| | 1931 July–1931 October | low number of daily measurements | -50 nT |
| | 1952 January–1952 October | new instruments | -20 nT |
| | 1952 November–1953 August | correction of the H series to the reference level in 1953 September | -20 nT + 5.4 nT = -15 nT |
| *AdB* | 1953 September–1959 August | correction of the H series to the reference level in 1959 August | 0 nT |
| | 1959 September–1969 April | correction of the H series to the reference level in 1968 April | +18 nT |
| | 1982 February | an outlier, source is unknown (low number of daily measurements?) | +30 nT |
| | 2001 October– December | an outlier, low number of daily measurements | +25 nT |
| | | I (1866-1941) | |
| *C-AdB* | 1866 June–1876 August | new instrument | +12' |
| | 1876 September–1922 October | new needles | +6' |
| | 1922 November–1928 October | this time interval is used as a reference level (see Sec. 3.2) | 0' |
| | 1928 November–1930 December | new needles | +5' |
| *AdB* | 1931 January–1938 December | relocation and degradation of the instruments | d.n.u. |
| | 1939 January–1940 May | data from unknown source | d.n.u. |
| | | Z (1951-2015) | |
| | 1951 October–1952 December | new instrument | -28 nT |
| | 1953 January–1954 May | unknown source for HB (no logbooks) | -2 nT |
| | 1954 June–1955 October | unknown source for HB (no logbooks) | -18 nT |
| | 1955 November–1960 October | unknown source for HB (no logbooks) | -29 nT |
| *AdB* | 1960 November–1971 August | unknown source for HB | -34 nT |
| | 1971 September–1973 November | unknown source for HB | -13 nT |
| | 1973 December–1989 October | this time interval is used as a reference level (see Sec. 3.3) | 0 nT |
| | 1989 November–2006 May | fast decrease; an annual cycle appears | +40nT |
| | 2006 June | changes from measurements to calculations (see Sec. 3.3) | -59 nT |
| | | D (2006-2015) | |
| *AdB* | 2006 January | new instrument | -7.3' |
| | | F (2006-2015) | |
| *AdB* | 2006 June | new instrument | +22 nT |
| | 2007 May | new instrument | -73 nT |