# Peer review of "Homogenization of the historical series from the Coimbra Magnetic Observatory, Portugal"

_Earth System Science Data, 2020_

## Referee Comment (RC1) · Mioara Mandea (Referee) · 12 Nov 2020

I am in the inevitability to recommend some revisions, as this paper has to be re-worked to attain an acceptable level for publication in ESSD. The paper is itself already very lengthy (without considering the SM), the data and the method are not particularly new, but obtaining a long series of magnetic measurements is important. Lots of information are given in the manuscript and they might be better deliver and summarized in forms of Tables. A physical interpretation of the results will bring more interest to this data-driven manuscript.

Specific comments

This manuscript is based on a huge effort to gather magnetic data from Coimbra observatory and investigate their quality in order to obtain a consistent long series of data.

The authors propose to homogenize the available magnetic data until 2015. Why not until present day, supposing data are available for mid-2020?

It is not clear how different noise contributions are assessed. For example, the "urban electromagnetic noise level" is noted, but no information about its value and evolution is provided.

The long series of D and/or Y component can be used to investigate the SV behavior and geomagnetic jerks. A comparison with long series of D measurements could be used to show how COI are data in investigating these events. Somehow Figure 1, has to be the last figure of the paper (with all corrections applied). This figure needs after to be discussed in details.

---

## Referee Comment (RC2) · Vadim Soldatov (Referee) · 26 Nov 2020

This is a continuation work extending homogeneity analysis of historical geomagnetic data series to cover all the magnetic field components. Such long historical records are extremely rare and most valuable, especially the very first years of operation, when Europe (and the world) had 3-5 observatories only. This work is a great effort to introduce these data into scientific circulation. The resulting time series will definitely help other researchers in studying the geomagnetic activity in pre-satellite era. The complete data sets are accessible as Excel files, both original and corrected. The format is clear, comprehensible and undoubtedly usable. Although my personal preference is to use more specific formats, such as netCDF or HDF. Plain ASCII text would do as well, as it allows for the direct comparizon of two corresponding files line by line. The

method chosen is appropriate for the purpose declared. The paper is quite detailed, not to say somewhat lengthy, but hopefully, these details would help potential users of the data in resolving any ambiguities they might encounter. The overall structure of the paper is clear and corresponds to the logic of the study.

———————————————————

---

## Referee Comment (RC3) · Mioara Mandea (Referee) · 8 Dec 2020

Dear authors

Many thanks for your detailed answers.

Best wishes,

---

## Author Response (AR1)

General comments on the revised manuscript:

The text of the manuscript and the Supplement were edited accordingly to the comments of two Reviewers (detailed replies to both Reviewers are below).

The main text was shortened with details about the instruments, when essential for future users of the homogenized data sets, moved to the updated Tables. Some English corrections are introduces. The Tables are updated: now all the metadata for the H, I and Z components are summarized in Tables 1-3 of the main manuscript. The old Figure 1 is transformed into two: new Fig. 1 and new Fig. 9 (as proposed by Dr. M. Mandea).

All changes are marked using the MS Words Markups except for the Tables which are shown in the updated form only. Please see the manuscript version with the mark-ups below.

The Supplement is changed accordingly to the changes of the main text: old Table 1 of the main text is moved to the Supplement, the old Tables S1-S3 are combined with the old Table 2 of the main text resulting in the new Tables 1-3.

The datasets are updated accordingly to the Dr. V. Soldatov comments: the files in the ASCII formats are added.

Reply to Dr. Mioara Mandea (essd-2020-317-RC1)

First of all we would like to thank Dr. Mioara Mandea for her useful comments.

Unfortunately we can't follow some of the suggestions (*"A physical interpretation of the results will bring more interest to this data driven manuscript"* and *"The long series of D and/or Y component can be used to investigate the SV behavior and geomagnetic jerks. A comparison with long series of D measurements could be used to show how COI are data in investigating these events."*). The main reason is that it is outside the ESSD journal policy which says *" Any interpretation of data is outside the scope of regular articles"*. Thus in this manuscript we focus on presenting new (homogenized) data sets that, we hope, will be used later on by other researchers to study geomagnetic field variations in the European region. Besides, we must note that the geomagnetic jerks were already studied in our 1[st] paper (*Morozova, A.L., Ribeiro, P., Pais, M. A.: Correction of artificial jumps in the historical geomagnetic measurements of Coimbra Observatory, Portugal, Ann. Geophys., 32, 19-40, doi:10.5194/angeo-32-19-2014, 2014.*) dedicated to the homogenization of the series of the D element. Since the Y element, as is shown as well in our present manuscript, is very similar to D and was always calculated from other measured elements, it has no additional value for a study of the jerks.

Below we present replies to other comments:

***Lots of information are given in the manuscript and they might be better deliver and summarized in forms of Tables.***
We combined the information from the main text Table 2 and Tables S1-S3 from the supplement to make four new tables (Tables 1-4 in the revised manuscript) with the metadata and the correction values for the H (Tab. 1-2), I (Tab. 3) and Z (Tab. 4) elements updating them also with the information from the main text. Also all details about the instruments (specific names and numbers and installation options) are moved from the main text to the Supplement (Table S3). Furthermore, the Table 1 from the original manuscript is also moved to the Supplement (Tab. TS2 in the updated Supplement). Other Tables in the supplement are not changed but renumbered accordingly to the main text changes.

***The authors propose to homogenize the available magnetic data until 2015. Why not until present day, supposing data are available for mid-2020?***
The set of the instruments installed in 2006-2007 is still in use at the COI Observatory. This means that, as we mention in our manuscript (sec. 1): "*the addition to the corrected series of measurements done after December 2015 will not affect their homogeneity*". Thus, anyone can download COI data for 2016-2017 from, e.g., WDC open data base and add them to the homogenized series presented in our work without any correction or treatment (corresponding link is added to the revised text – sec. 5). The data for 2018 and 2019 will be soon uploaded to WDC.

***It is not clear how different noise contributions are assessed. For example, the "urban electromagnetic noise level" is noted, but no information about its value and evolution is provided***.

In the presented manuscript we paid our attention only to changes of the baseline of the geomagnetic field elements. No treatment for the noise was done. The level of the urban electromagnetic noise could be devised from the variability of the data, e.g., using the month-to-month time derivative, as we mention in sec. 3.1 of the manuscript, or the standard deviation, comparing a more perturbed period vs a less perturbed period. Although, one must keep in mind that a larger part of hourly and daily noise is averaged out by the calculation of the monthly means. Corresponding sentence is added to the revised manuscript – sec. 5.

***Somehow Figure 1, has to be the last figure of the paper (with all corrections applied). This figure needs after to be discussed in details.***
Figure 1 from the original manuscript is split into Fig. 1 and 9 of the revised manuscript. Fig. 1 of the revised manuscript shows the COI original (observed) series and the COV-OBS model prediction. Figure 9 of the revised manuscript (now it is the last figure of the main text) shows COI observed and corrected series (final correction to the level of 2015). Since all the differences between the observed and corrected COI series result from the corrections described in detail in the main text (Sec. 3-4) we see no need to discuss once again the differences between the original and corrected series of the geomagnetic elements.

Also, according to the comments of another reviewer, the COI data are now available in the plain ASCII format in addition to the originally uploaded XLSX files.

Reply to Dr. Vadim Soldatov (essd-2020-317-RC2)

We would like to thank Dr. Vadim Soldatov for appreciation of our work and for providing us with a few practical suggestions

As was suggested by the reviewer, ("*The complete data sets are accessible as Excel files, both original and corrected. The format is clear, comprehensible and undoubtedly usable. Although my personal preference is to use more specific formats, such as netCDF or HDF. Plain ASCII text would do as well, as it allows for the direct comparison of two corresponding files line by line*."), we uploaded the data in the plain ASCII format to https://doi.org/10.5281/zenodo.4308022 (original data) and to https://doi.org/10.5281/zenodo.4308036 (homogenized data).

The reviewer also mentioned that the manuscript is "*somewhat lengthy*". In the revised manuscript we summarized the COI history in Tables 1-3 of the revised manuscript leaving as well many details for those potential users that will need to resolve ambiguities they might encounter using the COI historical data (e.g., if they will need to take care of the urban noise).

Thus, we combined the information from the main text Table 2 and Tables S1-S3 from the supplement to make four new tables (Tables 1-4 in the revised manuscript) with the metadata and the correction values for the H (Tab. 1), I (Tab. 2) and Z (Tab. 3) elements updating them also with the information from the main text. Also all details about the instruments (specific names and numbers and installation options) are moved from the main text to the Supplement (Table S3). Also, the Table 1 from the original manuscript is also moved to the Supplement (Tab. TS2 in the updated Supplement). Other Tables in the supplement are not changed but renumbered accordingly to the main text changes.

Accordingly to the comments of another reviewer, Figure 1 from the original manuscript is transformed into Fig. 1 and 9 of the revised manuscript. Fig. 1 of the revised manuscript shows the COI original (observed) series and the COV-OBS model prediction. Figure 9 of the revised manuscript (now it is the last figure of the main text) shows COI observed and corrected series (final correction to the level of 2015).

[revised manuscript text omitted]

---

## Editor Decision (ED1)

Dear Anna and co-authors,

many thanks for addressing the referee comments and my editorial remarks mostly. It seems, however, that you have missed some suggestions (1, 2) and I would like to ask you to including the DOI of the kp index instead of an outdated URL (3).

(1)

Unfortunately, data availability section still seems unchanged for me. Due to the versioning workflow of Zenodo that provides prominent links to all versions of a data via the Zenodo DOI Landing Page, there is no need to provide all four DOIs (of versions 1 and 2 for the two data sets). Instead, it is fully sufficient to only include the DOIs of version 2 (and reduced potential confusion for the readers). May I please ask you to change the data availability section

From

*"The COI geomagnetic field elements are available via the following addresses: COI original data – XLSX: https://doi.org/10.5281/zenodo.4122066 (Ribeiro et al., 2020a) and ASCII: https://doi.org/10.5281/zenodo.4308022 (Ribeiro et al., 2020b); COI homogenized data –XLSX: https://doi.org/10.5281/zenodo.4122289 (Morozova et al., 2020a) and ASCII: https://doi.org/10.5281/zenodo.4308036 (Morozova et al., 2020b)."*

To

"The COI geomagnetic field elements are available (in XLSX and ASCII formats) via the following addresses: COI original data – https://doi.org/10.5281/zenodo.4122066 (Ribeiro et al., 2020); COI homogenized data – https://doi.org/10.5281/zenodo.4308036 (Morozova et al., 2020)."

(2)

Line 486 (manuscript version3): please change "doi:" to https://doi.org/"

(3)

line 491: I have seen that you are using the old URL to the kp Index. This URL is still resolving, but only linking to another page. Moreover, the kp Index will be published with a DOI within the next weeks. Even thought the DOI of the kp Index is not yet resolving, I would be happy if you (1) included it already in the manuscript and (2) added the following reference to the References section:

Matzka, Jürgen; Bronkalla, Oliver; Tornow, Katrin; Elger, Kirsten; Stolle, Claudia (2021): Geomagnetic Kp index. V. 1.0. GFZ Data Services. https://doi.org/10.5880/Kp.0001

The change of the last sentence of the data availability section would then be (lines 491-492)

From

"The global Kp index can be downloaded from http://www-app3.gfz-potsdam.de/kp_index/index.html or http://isgi.cetp.ipsl.fr/des_kp_ind.html"

to

"The global Kp index can be downloaded from https://doi.org/10.5880/Kp.0001 (Matzka et al., 2021, see https://www.gfz-potsdam.de/kp-index/ for a general overview) or http://isgi.cetp.ipsl.fr/des_kp_ind.html"

(I will make sure that the DOI of the Kp index is online when your article will be published)

Many thanks and best regards, Kirsten

---

## Author Response (AR2)

Dear Kirsten,

I updated the "Data availability" section and the Reference list as you suggested.

Thank you very much for providing a new reference to the Kp index.

All the best,
Anna Morozova